# Myeloperoxidase-Oxidized LDL Activates Human Aortic Endothelial Cells through the LOX-1 Scavenger Receptor

**DOI:** 10.3390/ijms23052837

**Published:** 2022-03-04

**Authors:** Layal El-Hajjar, Judy Hindieh, Rana Andraos, Marwan El-Sabban, Jalil Daher

**Affiliations:** 1Department of Anatomy, Cell Biology and Physiological Sciences, Faculty of Medicine, American University of Beirut, Beirut 1107 2020, Lebanon; lh85@aub.edu.lb (L.E.-H.); me00@aub.edu.lb (M.E.-S.); 2Department of Biology, Faculty of Arts and Sciences, University of Balamand, Kalhat, Tripoli P.O. Box 100, Lebanon; judy.hindieh@std.balamand.edu.lb (J.H.); rana.andraos@fty.balamand.edu.lb (R.A.)

**Keywords:** atherosclerosis, Mox-LDL, endothelial dysfunction, LOX-1, inflammation, tubulogenesis

## Abstract

Cardiovascular disease as a result of atherosclerosis is a leading cause of death worldwide. Atherosclerosis is primarily caused by the dysfunction of vascular endothelial cells and the subendothelial accumulation of oxidized forms of low-density lipoprotein (LDL). Early observations have linked oxidized LDL effects in atherogenesis to the lectin-like oxidized low-density lipoprotein receptor-1 (LOX-1) scavenger receptor. It was shown that LOX-1 is upregulated by many inflammatory mediators and proatherogenic stimuli including cytokines, reactive oxygen species (ROS), hemodynamic blood flow, high blood sugar levels and, most importantly, modified forms of LDL. Oxidized LDL signaling pathways in atherosclerosis were first explored using LDL that is oxidized by copper (Cuox-LDL). In our study, we used a more physiologically relevant model of LDL oxidation and showed, for the first time, that myeloperoxidase oxidized LDL (Mox-LDL) may affect human aortic endothelial cell (HAEC) function through the LOX-1 scavenger receptor. We report that Mox-LDL increases the expression of its own LOX-1 receptor in HAECs, enhancing inflammation and simultaneously decreasing tubulogenesis in the cells. We hypothesize that Mox-LDL drives endothelial dysfunction (ED) through LOX-1 which provides an initial hint to the pathways that are initiated by Mox-LDL during ED and the progression of atherosclerosis.

## 1. Introduction

Atherosclerosis is a clinical condition for which multiple genetic and environmental causal factors have been proposed. A key process in the development of the disease is the accumulation of foam cells, macrophages that have engulfed large amounts of modified low-density lipoprotein (LDL) particles, which will result in the thickening of the arterial wall [1,2]. Myeloperoxidase (MPO) is a protein secreted by immune cells and a major physiological player in the generation of modified LDL molecules [3,4] via the production of hypochlorous acid (HOCl) from H_2_O_2_ and chloride [5]. LDL molecules that are modified by the latter enzymatic products were shown to be expressed in atherosclerotic lesions, both in vascular cells and in extracellular spaces [3]. Many studies have reported that patients with MPO-deficiency have a reduced risk of cardiovascular disease [4,6]. Other studies have also shown a correlation between myeloperoxidase oxidized LDL (Mox-LDL) and different levels of endothelial cell dysfunction (ED) [7]. As far as atherosclerosis is concerned, ED and circulating endothelial cells are associated with plaque rupture, stroke, and myocardial infarction [8]. It is already known that dysfunctional endothelial cells have reduced motility, which could affect their ability to undergo angiogenesis, an essential physiological mechanism seen in health and diseases including atherosclerosis [9,10,11,12]. On that same note, many studies have reported that dysfunction in endothelial cells, that leads to a disorganization in their regular cobblestone arrangement and an increase in their permeability, is considered as a causative condition that may predispose to atherosclerosis [13,14,15]. It has been also described that endothelial cell denudation is frequently observed at atheroma plaque locations and a failure to locally regenerate the endothelium is suspected to be deleterious in the context of atherosclerosis [16]. Furthermore, many post-acute myocardial infarction interventions were seen to be inefficient due to the inability of the endothelium to regenerate and heal the denuded regions [17]. As far as Mox-LDL is concerned, we had previously reported [18,19] that Mox-LDL induces ED by reducing both the fibrinolytic capacity of endothelial cells as well as their ability to migrate and undergo wound healing in vitro. Thus, our previous results suggested that high Mox-LDL levels in patients would impair the potential of endothelial cells to cope with the damaged endothelium, negatively contributing to the progression of the atheroma plaque. Despite the significance of the finding, we did not link this observation to any signaling transduction pathway, including the receptor that can bind to Mox-LDL and mediate its effects in the cell.

Initially, LOX-1 was identified as the major receptor for copper oxidized LDL (Cuox-LDL) in endothelial cells. LOX-1 is a type II membrane protein with a typical C-type lectin structure at the extracellular C-terminus. It is converted into a soluble form by proteolytic cleavage at the extracellular juxtamembrane region [20]. The lectin domain of LOX-1 is the functional domain that binds to its respective ligands, including Cuox-LDL; the latter binds to the receptor through the C-terminal end residues and several conserved positively charged residues spanning the lectin domain [21]. LOX-1 activation by Cuox-LDL leads to an increase in reactive oxygen species’ (ROS) production and the induction of nuclear factor-kappaB (NF-kB), subsequently upregulating the expression of adhesion molecules on the surface of endothelial cells. In vitro, LOX-1 is upregulated by many inflammatory factors including oxidative stress, cytokines, and Cuox-LDL. In vivo, LOX-1 expression is shown to be increased due to multiple proatherogenic stimuli such as hemodynamic blood flow, high blood sugar levels, hypertension, and hyperlipidemia. Immunohistochemical analyses have also confirmed that LOX-1 is upregulated in atherosclerotic lesions [22,23,24]. 

Yet, there is absolutely no information with respect to the role of the Mox-LDL receptor in cell models of atherosclerosis. Therefore, a dissection of the molecules involved in signal transmission of Mox-LDL effects is crucial in order to better understand the importance of this particular type of modified LDL in ED and atherogenesis. Accordingly, our hypothesis was the following: Mox-LDL drives ED through the LOX-1 scavenger receptor by mainly increasing the inflammatory state in endothelial cells. In the present work, we propose an initial model that delineates the first step in the molecular mechanism pertaining to the role of Mox-LDL in the progression of the atherosclerotic disease.

## 2. Results

### 2.1. Validation of LOX-1 Knockdown in HAEC

LOX-1 expression was assessed in HAECs that were either left untreated or transfected with negative control or LOX-1 siRNA. Figure 1A shows representative images of HAECs in those conditions; LOX-1 knockdown had no effect on HAECs morphology. Besides, LOX-1 expression decreased significantly after transfection (*p* < 0.05) as assessed by qPCR (Figure 1B). These results validate that LOX-1 knockdown was successfully achieved in HAECs. 

### 2.2. The Effect of Mox-LDL and siRNA Treatment on Cell Morphology

It was already reported that Mox-LDL does not affect cell morphology, viability, or death in human umbilical vein endothelial cells (HUVECs) [19]. In order to confirm that treatment with Mox-LDL and siRNA had no effect on HAECs morphology, cells that were transfected with either negative control or LOX-1 siRNA and treated with mock medium (CTL) or Mox-LDL (MOXLDL) medium were then visualized under bright field microscopy. As expected, HAECs showed no change in cell morphology in all treatment conditions. However, treatment with Mox-LDL relatively affected HAECs’ confluency compared to control (Figure 2).

### 2.3. The Effect of Mox-LDL on LOX-1 Expression

Many studies reported that the LOX-1 scavenger receptor is expressed on endothelial cells and is responsible for binding to oxidized LDL particles, ED, and the subsequent progression of atherosclerotic disease. It has been also shown that Cuox-LDL upregulates LOX-1 expression by binding to this receptor which leads to an increase in vascular dysfunction [25,26]. LOX-1 expression was assessed in HAECs that were treated with mock medium (CTL) or Mox-LDL (MOXLDL). In addition, in order to evaluate the possibility of Mox-LDL exerting its effect on LOX-1 in a positive feedback loop, cells were either transfected with negative control or LOX-1 siRNA and then treated with mock medium (CTL) or Mox-LDL (MOXLDL) and assessed for LOX-1 expression. Figure 3 shows that LOX-1 was highly upregulated (*p* < 0.001) in Mox-LDL-treated HAECs that were transfected with negative control siRNA. On the other hand, LOX-1 knockdown cells increased their expression of LOX-1 after Mox-LDL treatment, but still to a lesser extent as compared to their counterpart control cells. These results validate that Mox-LDL increases the expression of the LOX-1 scavenger receptor, and its mode of action might be in part related to an interaction with the scavenger receptor itself.

### 2.4. The Effect of Mox-LDL Treatment and LOX-1 Knockdown on IL-8 and NF-kB Expression

Chemokines, such as IL-8, contribute to inflammation and dysfunction in endothelial cells and are initiated through activation of the classical NF-kB pathway. In addition, it has been reported that IL-8 is upregulated in endothelial cells when treated with physiological concentrations of Mox-LDL [27,28]. Thus, the expression of IL-8 was first assessed in HAECs using immunofluorescence and was used as a readout in order to study the effect of Mox-LDL in association with LOX-1 expression. As predicted, immunofluorescence analysis showed that treatment with physiological concentrations of Mox-LDL increased the expression of IL-8 in cells transfected with negative control siRNA. However, HAECs that were transfected with LOX-1 siRNA and treated with Mox-LDL did not express IL-8 to the same extent as in their counterpart control cells, as shown in Figure 4A, as the expression of IL-8 was restricted to only some discrete parts of the cells as compared to the control condition. To verify the effect of LOX-1 knockdown on both IL-8 and NF-kB, their expression was assessed using qPCR and Western blotting analyses, respectively. In line with the immunofluorescence results, the level of mRNA expression of IL-8 was significantly increased when HAECs were treated with Mox-LDL (MOXLDL) compared to cells that were treated with mock medium (CTL) (*p* < 0.05). Similarly, LOX-1 knockdown HAECs did not upregulate IL-8 expression to the same extent as in the cells that were transfected with a negative control siRNA as shown in Figure 4B. As for NF-kB expression, cultured HAECs in normal conditions showed very minimal NF-kB activation. However, treatment of cells with Mox-LDL markedly enhanced the activation of NF-kB in comparison with untreated cells (*p* < 0.01). On the other hand, treatment of cells with LOX-1 siRNA before the cells were exposed to Mox-LDL significantly (*p* < 0.01) reduced Mox-LDL–mediated activation of NF-kB (Figure 5). Overall, these data suggest that Mox-LDL exerts its pro-inflammatory effects by binding to the LOX-1 scavenger receptor.

### 2.5. The Effect of Mox-LDL and LOX-1 Knockdown on IL-8 Secretion

To further determine the outcome of LOX-1 silencing on the capacity of Mox-LDL to activate inflammation in HAECs, we characterized the IL-8 pro-inflammatory cytokine secretion profiles of Mox-LDL-treated (MOXLDL) or untreated HAECs (CTL) that were either transfected with negative control siRNA or LOX-1 siRNA. Analysis of cytokine release profiles clearly demonstrated that Mox-LDL significantly (*p* < 0.001) increases the secretion of IL-8 in HAECs (Figure 6). Remarkably, LOX-1-silenced cells that were treated with Mox-LDL exhibited significantly (*p* < 0.001) reduced secretion levels of IL-8 when compared to cells that were treated with the negative control siRNA (Figure 6). These results were in accordance with immunofluorescence and qPCR analyses and clearly reflected the importance of the LOX-1 receptor in the Mox-LDL-mediated pro-inflammatory pathway.

### 2.6. The Effect of Mox-LDL Treatment and LOX-1 Knockdown on Tubulogenesis

A preponderance of studies has linked LOX-1 to angiogenesis in both cancer and atherosclerosis models. It has been equally reported that Mox-LDL interferes with cell motility and wound healing in endothelial cells [19,29]. Thus, in order to investigate the dual effect of Mox-LDL treatment and LOX-1 silencing on HAECs’ ability to build a vascular bed network, an in vitro tubulogenesis assay was performed. HAECs that were treated with mock medium (CTL) formed a characteristic capillary-like network, as expected. Conversely, LOX-1 knockdown and Mox-LDL treatment decreased to a large extent the ability of HAECs to build vascular projections compared to control cells, whereby the effect was additive and the HAECs that were both treated with Mox-LDL (MOXLDL) and knocked down for LOX-1 demonstrated the largest degree of inability to undergo tubulogenesis (Figure 7A). Quantification analysis showed that this decrease was significant (*p* < 0.001) and the reduction in tubule formation was in the range of 40 to 60% in all treatment conditions: negative control siRNA; MOXLDL, LOX-1 siRNA; CTL, LOX-1 siRNA; MOXLDL whenever we compared them to the negative control siRNA; CTL condition; of note, Mox-LDL with LOX-1 siRNA treatments exhibited the highest reduction in tubulogenesis (Figure 7B).

## 3. Discussion

In our study, we investigated, for the first time, the interaction between Mox-LDL and LOX-1 in endothelial cells. Our data indicate that Mox-LDL may affect HAEC function through the LOX-1 scavenger receptor. Our results must be discussed in light of previously published studies on the effects of oxidized LDL in ED, where authors explored oxidized LDL signaling pathways related to inflammation and atherogenesis, using the Cuox-LDL model [30]. Conversely, very little is known about the Mox-LDL molecular pathways and processes in the context of atherosclerosis. LDL that is modified by MPO is the more physiologically relevant model of LDL oxidation; in fact, Daugherty et al. and others conducted many immunohistochemical analyses that have confirmed the presence of the MPO enzyme with some of its modified amino acids derivatives (in the ApoB100 moiety of LDL), such as chlorotyrosine, within the atheroma plaques of patients with the disease [5,31,32]. Therefore, in the present work, we aimed to briefly examine the role of Mox-LDL in ED using both the HAEC and Mox-LDL pathophysiological models.

We had previously reported that Mox-LDL treatment does not change the morphology or reduce the viability in both HUVECs and HAECs. Nevertheless, we have documented that Mox-LDL might interfere with endothelial cell behavior, mainly cell adhesion [19,33]. Those data come in line with the results of our present study where we have confirmed that physiological concentrations of Mox-LDL do not affect cell morphology but cell density only, an observation that could mostly be related to a decrease in cell adhesion after treatment. Overall, these results are in contrast with previous reports that pointed to a strong effect of Cuox-LDL on cell morphology and on reducing the viability of vascular endothelial cells [34,35]. This discrepancy could be principally related to the model for LDL oxidation or the treatment concentration used in those studies. The oxidation model and the concentrations that were used in our study are both pathophysiologically relevant and they reflect what could happen in in vivo environments. On that same note, it is imperative to state here that the widely used Cuox-LDL model requires very high concentrations that are unlikely to be available in vivo in order to sufficiently generate observable oxidized LDL concentrations.

LOX-1, a lectin-like receptor that has been proposed as a receptor for oxidized LDL, is expressed on endothelial cells and has been identified in the atherosclerotic arteries of human and several animal models. In vivo, this receptor was shown to be induced by oxidized LDL itself, as well as many pro-atherogenic factors such as shear stress, diabetes, dyslipidemia, endothelin and angiotensin II. In vitro, LOX-1 is upregulated by hemodynamic stimuli and a multitude of inflammatory cytokines, and by the copper oxidized form of LDL. Since LOX-1 binds to a variety of ligands that are crucial in the pathogenesis of atherosclerosis, it was proposed as an exciting target for drug therapy [20,23,36]. The latter drugs may include naturally occurring antioxidants with anti-inflammatory functions that have been demonstrated to inhibit LOX-1 expression and activity in vascular cells [37]. Food that is rich in polyphenols and flavonoids was shown to have strong antioxidant and anti-inflammatory activity [38,39]. Indeed, procyanidins were reported to significantly decrease oxidized LDL uptake in LOX-1 expressing cells in vitro by inhibiting the binding of the scavenger receptor to its modified LDL ligand. In vivo, procyanidins also demonstrated atheroprotective properties in hypertensive rats fed with a high fat diet by suppressing lipid accumulation in the walls of blood vessels [40].

The LOX-1 scavenger receptor was previously shown to be expressed on the surface of bovine aortic endothelial cells in the intima of the normal bovine aorta. The receptor was also shown to be induced by Cuox-LDL in vitro in this animal cell model and its upregulation was dose- and time-dependent [41]. Those results are in contrast to another study which showed that neutralizing anti-LOX-1 antibodies was not able to interfere with oxidized LDL signal transduction; it is important to mention here that in the latter study, the authors used a different model of oxidized LDL generation as well as a different endothelial cell model, which was the human EA.hy926 cell line [18]. Here again, this disagreement could be partly related to the model for LDL oxidation, as well as the cell model that was used in order to produce the data. Interestingly, in the same study of Zouaoui et. al, the authors analyzed the effect of LDL that was modified by MPO on a battery of putative receptors that might be implicated in MPO-modified LDL signaling transduction pathways. The receptors were of the scavenger receptors’ family and included SRB, SRD, and SRF which were shown not to be expressed in the EA.hy926 endothelial cell model.

A preponderance of evidence has linked the LOX-1 scavenger receptor to ED, induction of inflammation, and initiation of atherosclerosis [25,26]. Cuox-LDL was shown to bind to LOX-1 receptor increasing its expression at the transcriptional level. Intracellularly, this will induce the activation of membrane-bound NADPH oxidase leading to a rapid increase in ROS generation and exacerbation of the inflammatory response [42]. Although enough evidence had been accumulated to link Cuox-LDL to its respective receptor LOX-1, no research at this level was conducted on endothelial cells, mainly the HAEC model, using Mox-LDL.

Yet again, the oxidation theory proposes that LDL oxidation is an early event in atherosclerosis and that oxidized LDL contributes to atherogenesis by triggering inflammation through its interaction with the main scavenger receptor, LOX-1, expressed on the surface of endothelial cells [30]. Accordingly, we aimed to investigate the molecular pathways that are promoted by Mox-LDL in HAECs by inhibiting the expression of the LOX-1 receptor and detecting the outcomes. The results showed that LOX-1 knockdown was able to interfere with Mox-LDL signal transduction, as monitored by IL-8 induction.

It has been previously reported that MoxLDL triggers inflammatory pathways in both macrophages and endothelial cells [7] and that Mox-LDL treatment was found to induce IL-8 release in endothelial cells in a dose dependent manner [43]. IL-8 plays a crucial role in atherogenesis where it acts as a chemoattractant to inflammatory cells and also to smooth muscle cells where it is involved in their migration to the intima and the formation of the fibrous cap [28,44]. Delporte et al. has also linked Mox-LDL to IL-8 production in endothelial cells [28]. Besides, several studies have shown that IL-8 is involved in regulating endothelial cell permeability and is related to the vascular dysfunction associated with many vascular diseases including atherosclerosis [26,45].

In our study, we reported that Mox-LDL treatment in HAECs greatly increases LOX-1 mRNA levels (~16-fold) when compared to control untreated cells. Similarly, LOX-1 was also induced after Mox-LDL treatment in LOX-1 knockdown cells, but still to a lesser extent as in their counterpart control cells, which proves that the Mox-LDL mode of action might be in part related to an interaction with the scavenger receptor itself. Thus, our results suggest that Mox-LDL binds to LOX-1, increasing transcriptional activation of LOX-1 mRNA synthesis, entering in a positive feedback loop that exacerbates endothelial cell dysfunction.

On the other hand, we have shown that upregulation of IL-8 by Mox-LDL in our model is mediated by LOX-1 and NF-kB activation, since inhibition of LOX-1 by its specific siRNA reduced the expression and activation of IL-8 and NF-kB respectively, which was shown to be elicited by Mox-LDL. Immunofluorescence, qPCR, ELISA and Western blot analyses that we have conducted showed that treatment of HAECs with physiological concentrations of Mox-LDL significantly increased the expression of IL-8 and the activation of NF-kB, and that LOX-1 silencing interfered with this inflammatory response. This also points to the important role that the scavenger receptor LOX-1 might play in initiating the inflammatory signaling pathway in endothelial cells when treated with pathophysiological concentrations of Mox-LDL. These observations come in line with what had been previously reported about the relationship between Cuox-LDL and LOX-1 in endothelial cell models of atherosclerosis. As far as LOX-1 is concerned, our results are in perfect agreement with many studies that have shown that this scavenger receptor plays an important role in inflamed microenvironments and atherosclerotic lesions, where it mediates endothelial cell activation when the latter is exposed to proinflammatory and proatherogenic stimuli [46].

Regarding the effect of Mox-LDL on the in vitro capacity of endothelial cells to undergo angiogenesis, we were the first to show that Mox-LDL induces significant changes in HAECs at this level. Thus, in our hands, physiological concentrations of Mox-LDL were able to interfere with tubule formation in HAECs, a property which is mainly associated with in vivo angiogenesis. Our results are once again similar to those that have been reported on the effect of Mox-LDL regarding in vitro tubulogenesis [19]. On the other hand, Cuox-LDL has been linked to contradictory effects that are related to angiogenesis-like endothelial growth, where it was shown that Cuox-LDL might play a proangiogenic role at low concentrations, yet an inhibitory effect at higher concentrations [34,47,48]. In our results, we revealed that Mox-LDL treatment tremendously decreased HAECs’ ability to build a vascular network. More importantly, this effect was exacerbated by LOX-1 silencing whereby cells that were both knocked down for LOX-1 and treated with Mox-LDL demonstrated the largest anti-angiogenic effect. Our data point to a dual role of Mox-LDL in relation to its direct and indirect anti- and pro-angiogenic effects, respectively, reflecting the complexity of the processes that regulate the growth of the plaque and the mechanisms of extra- and intra-neo-angiogenesis in vivo. Once again, it is important to mention that we used an oxidation agent and a cell model which are both considered to perfectly reflect what happens in vivo situations.

Recent studies have linked LOX-1 to angiogenic mechanisms in both atherosclerosis and cancer models. In atherosclerosis, LOX-1 has been found to promote angiogenic processes [30]. Regarding the onset of cancer, LOX-1 was shown to be upregulated in different tumors and has been linked to their progression and metastasis through the upregulation of VEGF, the activation of HIF-1alpha, and the induction MMP-9/MMP-2, subsequently leading to neo-angiogenesis [49]. Similarly, it was reported in one study that LOX-1 is over-expressed in stage III and IV of human prostatic adenocarcinomas [50]. Another study has also shown that LOX-1 expression correlates with the aggressiveness of human colon cancer in vitro [51]. These findings underline the importance of the LOX-1 scavenger receptor as a promising therapeutic target in anticancer treatment strategies as well as a potential aim for anti-atherosclerosis studies.

In summary, Mox-LDL upregulates the expression of its own receptor in HAECs. Mox-LDL also induces inflammation in HAECs via LOX-1 in concert with the upregulation of this receptor, and last, Mox-LDL and LOX-1 have a dual role in affecting in vitro angiogenesis in HAECs. These observations may have important implications with regard to Mox-LDL-driven ED. Overall, our results provide an updating knowledge of the role of LOX-1 and Mox-LDL in endothelial cell activation by giving initial insights into the scavenger receptor that binds to this type of modified LDL and the possible signaling transduction pathways that are promoted by it. In this study, we have ultimately revealed that Mox-LDL acts through the LOX-1 receptor, which is the same receptor for Cuox-LDL. More generally, and in the context of the altered angiogenesis phenotype, high oxidized LDL levels could also impact the progression of many pathologies, including cancer. Recently, a strong correlation between metabolic disorders and the progression of cancer has been demonstrated, directing new therapeutic strategies on novel targets including LOX-1 for its important role in angiogenesis and inflammation.

Finally, in light of the importance of oxidized LDL-LOX-1 signaling pathways in the onset and progression of atherosclerosis, anti-atherogenic strategies that target Mox-LDL-LOX-1 interaction could be promoted as an exciting and promising avenue in developing therapeutic agents to alleviate the atherosclerotic process in humans. Therefore, the LOX-1 receptor may represent an attractive therapeutic target for the prevention and management of atherosclerosis and its related diseases. On that same note, other inflammatory markers besides IL-8 are still essential to tackle when dissecting the Mox-LDL signaling pathways in detail; this might be considered as a limitation in the current study. Nonetheless, the results that we are reporting pave the way to ongoing research where the effect of LOX-1 knockdown is studied on a battery of inflammatory markers in order to better understand the mechanism of action of Mox-LDL. Further elucidation of signaling pathways and novel functions of LOX-1 and Mox-LDL will definitely advance our understanding of the role of both players in the pathogenesis of atherosclerosis.

## 4. Materials and Methods

### 4.1. Cell Culture 

Human aortic endothelial cells (HAEC) (kindly provided by Dr. Marwan El-Sabban, American University of Beirut) were cultured in EBM-2 Basal Medium supplemented with EGM-2 SingleQuots™ Kit supplements and growth factors (Lonza). Cells were incubated at 37 °C in a humidified incubator (95% air, 5% CO_2_).

### 4.2. Oxidation of LDL

The enzymatic oxidation of LDL was performed by mixing HCl (1 M, 8 µL), MPO (11.11 × 10^−6^ M, 45 µL), 1.6 mg LDL (pH 7.4, 0.8 mg/mL in PBS, 200 µL), and H_2_O_2_ (0.05 M, 40 µL). The volume was completed with PBS (pH 7.4) containing 1 g/L of EDTA to reach a final volume of 2 mL. The oxidant/lipoprotein molar ratio is 625:1 in this condition [52].

### 4.3. Transfection

20,000 cells/cm^2^ were transfected with 5  nM LOX-1 specific OLR1-siRNA (Silencer Select Validated siRNA ID s9843, Ambion Applied Biosystems, Austin, TX, USA) or with a non-target siRNA (Silencer Select Negative Control, Ambion Applied Biosystems, Austin, TX, USA) using Hiperfect Transfection Reagent (Qiagen, The Netherlands) as per the manufacturer’s protocol. Briefly, 30,000 cell/cm^2^ were seeded in 24-well plates on the day of transfection in 100 µL complete media containing FBS and antibiotic and incubated under normal conditions (37 °C and 5% CO_2_). Then, 37.5 ng of siRNA were diluted in 100 µL of serum-free medium, and 3 µL of transfection reagent were added and mixed by vortexing. Samples were incubated for 5–10 min at room temperature to allow the formation of transfection complexes. After the incubation, complexes were added drop-wise onto the cells and mixed gently to ensure uniform distribution of the transfection complexes. Cells were then incubated under normal growth conditions and monitored for gene silencing after 24 h of transfection before proceeding with subsequent treatments. 

### 4.4. Mox-LDL Treatment

Normal and LOX-1 knockdown HAECs were seeded in two 6-well plates 30,000 cell/cm^2^. One row of each plate was either treated with Mox-LDL (100 µg/mL) or left untreated. Cell morphology was monitored after treatment and images (10× magnification) were captured using inverted light microscopy.

### 4.5. RNA Extraction and Quantitative PCR

Total RNA was isolated from cells in culture using Nucleospin^®^ RNA II Kit (Machery-Nagel, USA). Total RNA concentrations and A260/A280 were measured using Nanodrop. First 1 μg of total RNA was reverse transcribed to cDNA using iScript™ cDNA Synthesis Kit (ThermoFisher, Vilnius, Lithuania). Quantitative PCR (qPCR) was performed using iQ SYBR Green Supermix in a CFX96 system (Bio-Rad Laboratories, Hercules, CA, USA) using the primers listed in Table 1. The standard cycling condition was 95 °C for 5 min, followed by 40 cycles of 95 °C for 10 s, 60 °C for 30 s and 72 °C for 30 s. The results were analyzed using SDS 2.3 relative quantification manager software. The comparative threshold cycles values were normalized for GAPDH reference genes. The qPCR was performed in triplicate to ensure quantitative accuracy. The 2-ΔΔCq method was applied to calculate the relative fold change in gene expression after normalization.

### 4.6. Immunofluorescence 

HAECs that were either treated with Mox-LDL (100 µg/mL) or left untreated, were seeded onto coverslips and fixed with 4% PFA at 4 ºC. Cells were washed with PBS (1X) 3 times. Then, they were blocked for 1 h with PBS (1X) and 3% normal goat serum (NGS) and incubated again for 3 h with interleukin-8 (IL-8) antibody (abcam, USA) prepared in PBS (1X) and 1% NGS. Cells were further incubated with Alexa fluor 488 secondary antibodies (Invitrogen, USA) prepared in PBS (1X) and 1% NGS for 1 h. Nuclei were counterstained with Hoechst 33,324 (Thermo Fisher, Molecular Probes, Eugene, OR, USA). Coverslips were mounted onto glass slides using Prolong Antifade (Molecular Probes) and slides were observed under the LSM 710 microscope at 40× magnification.

### 4.7. Western Blot Analysis

Protein extraction was performed using the Nuclear and Cytoplasmic Protein Extraction kit (Thermo Scientific, Pierce Biotechnology, IL, USA) according to the manufacturer’s instructions. The samples were separated using 10% sodium dodecyl sulfate-polyacrylamide gel electrophoresis and then transferred onto PVDF membranes (Bio-Rad, Hercules, CA, USA). Subsequently, the membrane was blocked in TBST (20 mM Tris base pH 7.6, 150 mM NaCl, 0.1% Tween-20) containing 5% non-fat milk for 4 h at room temperature. Then, the membrane was incubated with NF-kB (Invitrogen, Cat number MA5-15160) and Lamin B1 (Invitrogen, Cat number 33-2000) primary antibodies that were added to the blocking solution at 3 µg/mL, and the membrane was incubated at 4 °C overnight. After overnight incubation, the membrane was washed and then incubated with goat anti-rabbit immunoglobulin G HRP conjugated secondary antibodies (Invitrogen, Cat number G-21234) and goat anti-mouse immunoglobulin G HRP conjugated secondary antibodies (Invitrogen, Cat number PA1-86015) for an additional hour at room temperature. After washing three times with TBST (30 min each wash), proteins were detected by chemiluminescence and protein bands were analyzed by densitometry using ImageLab software (Bio-Rad 6.1, Hercules, CA, USA).

### 4.8. Enzyme-Linked Immunosorbent Assay (ELISA)

The supernatants from HAEC cultures, that were transfected with the negative control siRNA or LOX-1 siRNA and that were either left untreated or treated with Mox-LDL, were collected and stored at −80 °C for later cytokine analysis. IL-8 levels in the HAEC culture supernatants were measured using a commercially available sandwich ELISA kit (Invitrogen). The samples were processed according to the manufacturer’s instructions in duplicates and measured at 450 nm on a micro-plate reader (Biotek, Winooski, VT, USA). 

### 4.9. Tubulogenesis

The Matrigel tube formation assay, or tubulogenesis assay, was performed to assess the ability of HAECs to form tubules involved in vessel formation (angiogenesis). Tubulogenesis was initiated using HAECs that were transfected with a negative control or LOX-1 siRNA and that were either left untreated or treated with Mox-LDL (100 µg/mL) for a period of 24 h. Briefly, Matrigel Basement Membrane (BD, Biosciences; Cat number 356234) was allowed to polymerize after incubating it for 2 h in a 24-well plate at 37 °C. Afterwards, the plate containing the now polymerized Matrigel was incubated for a supplementary 1 h after further addition of 250 μL of complete EBM medium to each well. Then, 5 × 104 of HAECs (+negative control siRNA; LOX-1 siRNA/±Mox-LDL) were added to each well in duplicates. After an overnight incubation period, cells were inspected for tubule formation and pictures were taken from three fields of views using inverted light microscopy at 10× magnification. The tubule networks’ lengths were measured using the Image J program, averaged between the fields of views and duplicates, and expressed as a percentage of control untreated cells. Experiments were reproduced three times independently.

### 4.10. Statistical Analysis

Statistical analysis was performed using GraphPad Prism software (version 6.0; GraphPad Software, Inc., San Diego, CA, USA). Data are expressed as the mean ± standard error mean (SEM) and *p* < 0.05 was considered to show a statistically significant difference. The statistical significance study of the results of qPCR, the Western blot for the validation of LOX-1 knockdown, the analysis of the effect of MPO-oxidized LDL on LOX-1 expression, the effect of MPO-oxidized LDL and LOX-1 knockdown on IL-8 and NF-kB expression, as well as tubulogenesis, were conducted using one-way ANOVA followed by Tukey’s multiple comparison post hoc test. All the experiments were repeated three independent times. 

## Figures and Tables

**Figure 1 ijms-23-02837-f001:**
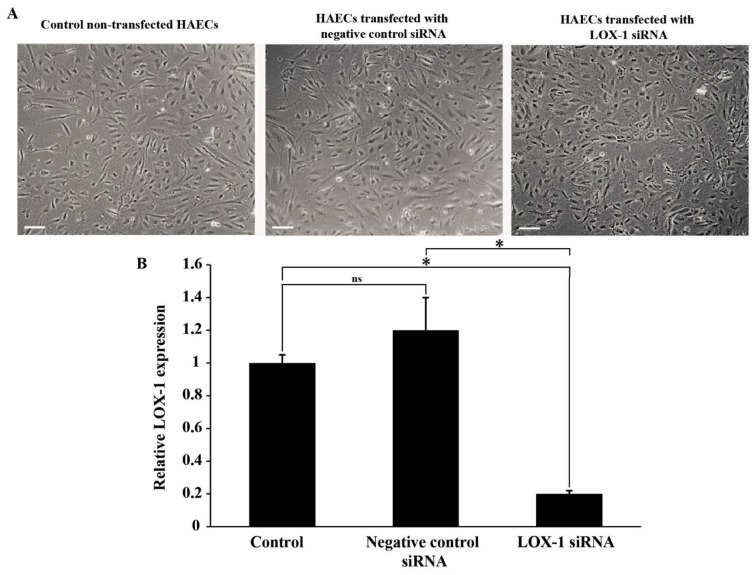
The effect of LOX-1 knockdown on HAECs. (**A**) Representative images of LOX-1 expression in HAECs that were left non-transfected or transfected with negative control or LOX-1 siRNA (scale bar: 200 μm); (**B**) Bar graph representing LOX-1 mRNA expression in HAECs that were left untreated (control) or treated with either negative control or LOX-1 siRNA, as detected by qPCR and normalized to GAPDH from three independent experiments. Data are presented as the mean ± SEM (*n* = 3) fold change in mRNA expression. Statistically significant differences were determined by one-way ANOVA followed by Tukey’s multiple comparison post hoc test (* *p* < 0.05, ns: not significant).

**Figure 2 ijms-23-02837-f002:**
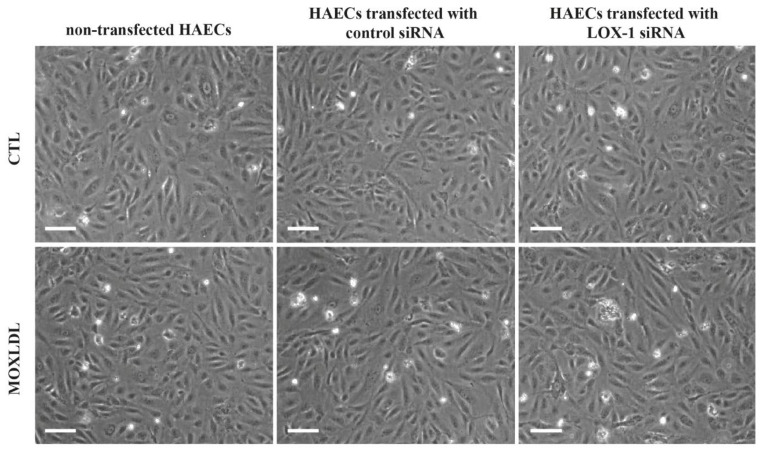
Representative bright field inverted microscopy images of HAECs under different treatment conditions. HAECs showed no change in morphology upon treatment with Mox-LDL, negative control siRNA or LOX-1 siRNA (scale bar: 200 μm).

**Figure 3 ijms-23-02837-f003:**
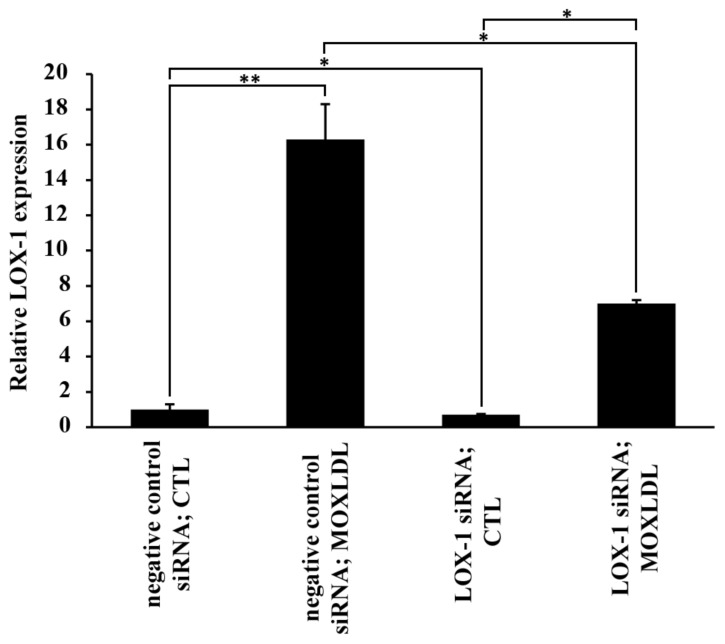
The effect of Mox-LDL on LOX-1 expression in HAECs. Bar graph representing LOX-1 mRNA expression in HAECs transfected with negative control or LOX-1 siRNA and then treated with mock medium (CTL) or Mox-LDL (MoxLDL), as detected by qPCR and normalized to GAPDH from three independent experiments. Data are presented as the mean ± SEM (*n* = 3) fold change in mRNA expression. Statistically significant differences were determined by one-way ANOVA followed by Tukey’s multiple comparison post hoc test (* *p* < 0.05; ** *p* < 0.01).

**Figure 4 ijms-23-02837-f004:**
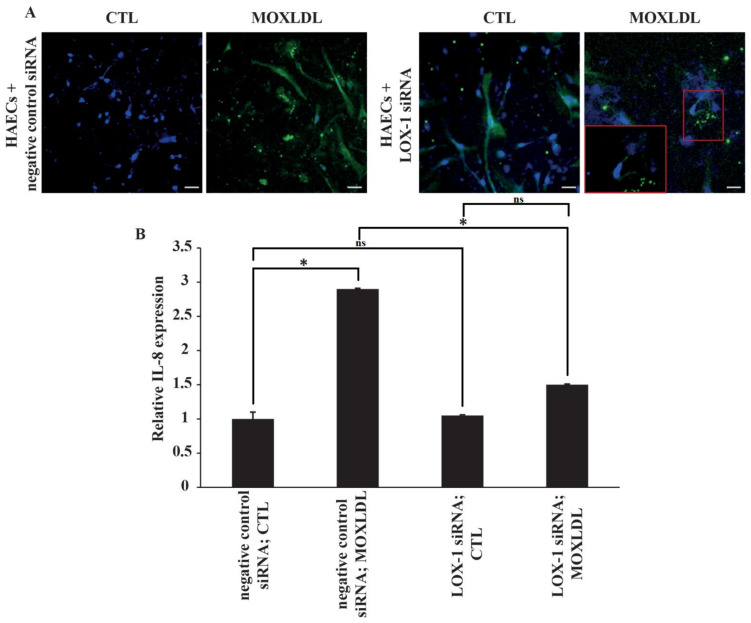
The effect of LOX-1 knockdown and Mox-LDL treatment on Il-8 expression. (**A**) Representative immunofluorescence images of IL-8 expression in HAECs that were transfected with either negative control or LOX-1 siRNA and subsequently treated with mock medium (CTL) or Mox-LDL (MOXLDL) for 24 h before visualization using fluorescence microscopy (scale bar: 50 μm); (**B**) Bar graph representing IL-8 mRNA expression in HAECs subjected to the same treatment conditions, as detected by qPCR and normalized to GAPDH from three independent experiments. Data are presented as the mean ± SEM (*n* = 3) fold change in mRNA expression. Statistically significant differences were determined by one-way ANOVA followed by Tukey’s multiple comparison post hoc test (* *p* < 0.05, ns: not significant).

**Figure 5 ijms-23-02837-f005:**
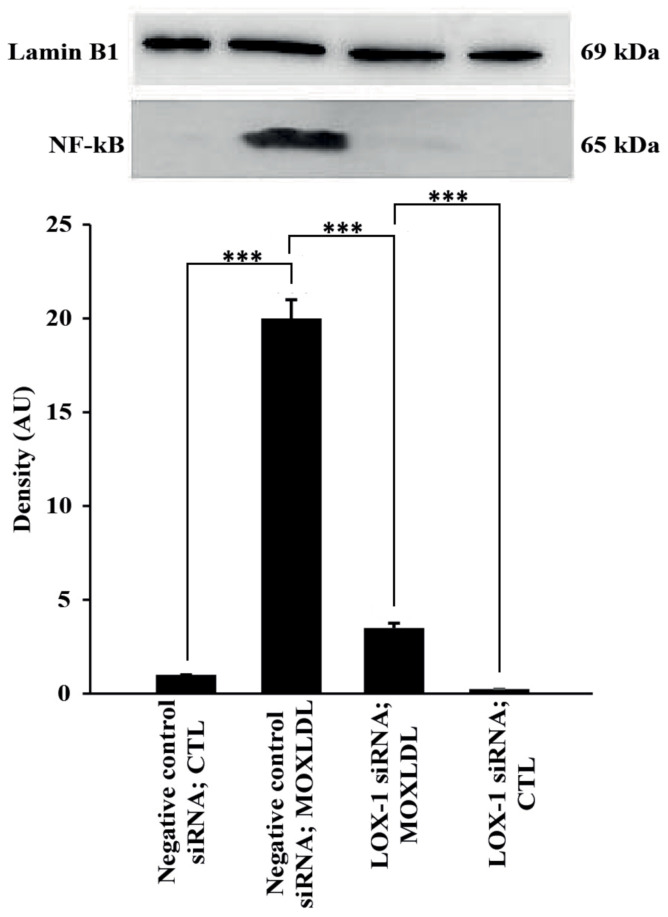
Identification of NF-kB protein by Western blot analysis. Incubation of HAECs with Mox-LDL leads to a significant increase in the activation of NF-kB (*p* < 0.01, negative control siRNA; MOXLDL vs. negative control siRNA; CTL). Treatment of HAECs with LOX-1 markedly decreases Mox-LDL-induced activation of NF-κB, negative control siRNA; MOXLDL vs. LOX-1 siRNA; MOXLDL). AU indicates arbitrary units. Data are representative of four separate experiments. Statistically significant differences were determined by one-way ANOVA followed by Tukey’s multiple comparison post hoc test (*** *p* < 0.001).

**Figure 6 ijms-23-02837-f006:**
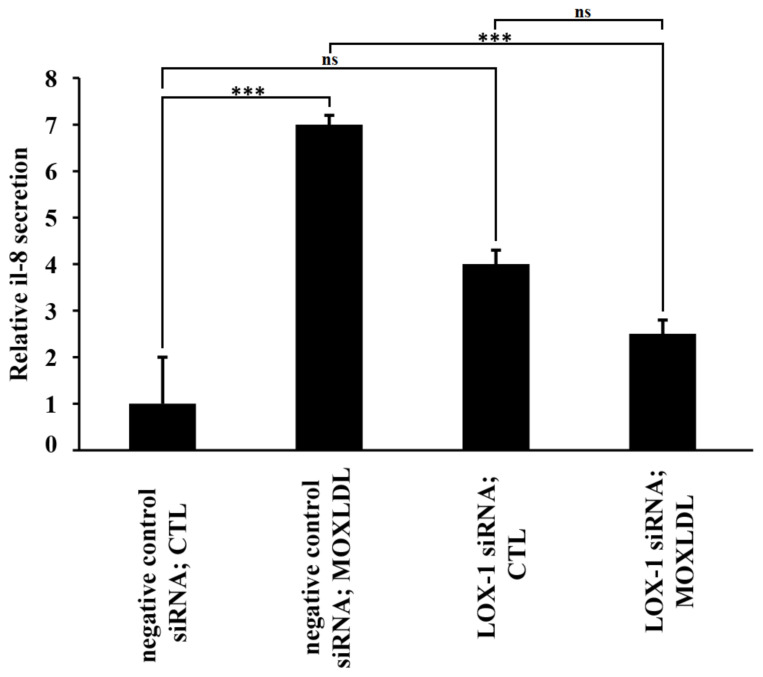
Measurement of IL-8 secreted by Mox-LDL treated HAECs that were transfected with LOX-1 siRNA. IL-8 levels (fold/control) in the culture supernatants of HAECs that were transfected with either negative control or LOX-1 siRNA and subsequently treated with mock medium (CTL) or Mox-LDL (MOXLDL) for 24 h, as measured by ELISA. Column bars represent mean values of three independent experiments. Error bars represent SEM. One-way ANOVA followed by Tukey’s multiple comparison post hoc test was used to calculate statistical significance. *** *p* < 0.001, ns: not significant.

**Figure 7 ijms-23-02837-f007:**
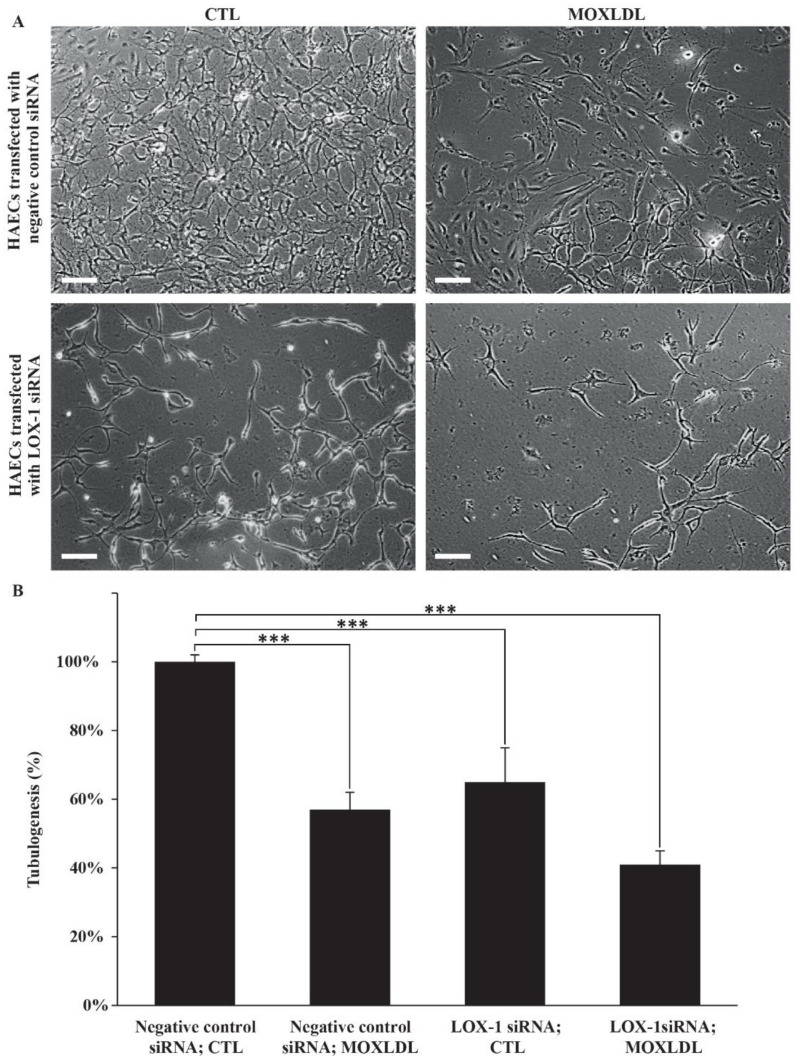
The effect of Mox-LDL treatment on HAECs’ tubulogenesis. (**A**) Representative fields of HAECs plated on Matrigel (scale bar: 200 μm). HAECs that were transfected with either negative control or LOX-1 siRNA were subsequently seeded on Matrigel supplemented with mock medium (CTL) or Mox-LDL (MOXLDL) and incubated overnight; (**B**) Quantification of tubule formation in HAECs. After overnight incubation in different treatment conditions, pictures were taken from multiple fields of view and the cumulative tubule length was assessed using the Image J program. The data are expressed as total tubule network length for each treatment condition normalized to the untreated control taken at 100%. Data are representative of three separate experiments. Statistically significant differences were determined by one-way ANOVA followed by Tukey’s multiple comparison post hoc test (*** *p* < 0.001).

**Table 1 ijms-23-02837-t001:** List of primers.

Gene	Primer Sequence
LOX-1	F:CCACCAGAATCTGAATCTCCAAGAA R:ACTTGGCATCCAAAGACAAGCAC
IL-8	F:GAGAGTGATTGAGAGTGGACCAC R:CACAACCCTCTGCACCCAGTTT
GAPDH	F:TGGTGCTCAGTGTAGCCCAG R:GGACCTGACCTGCCGTCTAG

## Data Availability

All data generated or analyzed during this study are included in this article.

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
