# Peer review of "Myeloperoxidase-Oxidized LDL Activates Human Aortic Endothelial Cells through the LOX-1 Scavenger Receptor"

_ijms, 2022, doi:10.3390/ijms23052837_

Round 1
Reviewer 1 Report
DearAuthors
The manuscrit here deals with the LOX-1 scavenger receptor and its interaction with bioendogenous compounds.
The topic is within the aim and scope of the SI and attractive for a broad range of audience. The text is correclty organized and the experimental part well detailed, however english language needs a revision for typos and grammar, also please add the statystical part at your in vivo data.
Resolution of images is good, I suggest to discuss a little bi the importance of this receptors in inflammation process, also in relation to other systems for example look at the following literature:"Phenolic Analysis and In Vitro Biological Activity of Red Wine, Pomace and Grape Seeds Oil Derived from Vitis vinifera L. cv. Montepulciano d’Abruzzo"; "Discovery of arginine-containing tripeptides as a new class of pancreatic lipase inhibitors"; "Exploring the Nutraceutical Potential of Dried Pepper Capsicum annuum L. on Market from Altino in Abruzzo Region".
Reviewer 2 Report
The authors stated that the interaction between Mox-LDL and LOX-1, and their role in affecting in vitro angiogenesis in HAECs. Some comments:
- In the abstract section, it is better for the authors to elaborate more on their research findings or conclusions rather than a background introduction.
- In all figures, it is advisable for the authors to mark which two groups of comparisons the significant difference (asterisk) refers to, so that the reader can easily distinguish between them. In Figure 1, no differences between the two control groups are preferably marked (e.g., ns), as this determines whether the transfection procedure affects the experimental results.
- Please note some of the superscript and subscript symbols. For example, H2O2, CO2, etc.
- In line 209, “As expected, HAECs showed no change in cell morphology in all treatment conditions. However, treatment with Mox-LDL reduced considerably HAECs count compared to control.” This description is not very accurate. It is recommended to list objective expressions, such as density, etc.
- The authors claim "we investigated, for the first time, the interaction between Mox-LDL and LOX-1 in endothelial cells", but I don't think the Fig.3 alone is sufficient proof. In other words, siRNA alone is not sufficient, but also overexpression or other correlation experimental evidence is needed. This also applies to several other conclusions, such as IL-8, and tubulogenesis.
Round 2
Reviewer 2 Report
no comments